# Interleukin Inhibitors in Cytokine Release Syndrome and Neurotoxicity Secondary to CAR-T Therapy

**DOI:** 10.3390/diseases10030041

**Published:** 2022-07-06

**Authors:** Puri Ferreros, Isabel Trapero

**Affiliations:** Nursing Department, Faculty of Nursing and Podiatry, University of Valencia, 46010 Valencia, Spain; isabel.trapero@uv.es

**Keywords:** CAR-T cells, cytokine release syndrome, neurotoxicity, interleukin inhibitors

## Abstract

Introduction: Chimeric antigen receptor T-cell (CAR-T) therapy is an innovative therapeutic option for addressing certain recurrent or refractory hematological malignancies. However, CAR-T cells also cause the release of pro-inflammatory cytokines that lead to life-threatening cytokine release syndrome and neurotoxicity. Objective: To study the efficacy of interleukin inhibitors in addressing cytokine release syndrome (CRS) and neurotoxicity secondary to CAR-T therapy. Methodology: The authors conducted a bibliographic review in which 10 articles were analyzed. These included cut-off studies, case reports, and clinical trials involving 11 cancer centers and up to 475 patients over 18 years of age. Results: Tocilizumab is the only interleukin inhibitor approved to address CRS secondary to CAR-T therapy due to its efficacy and safety. Other inhibitors, such as siltuximab and anakinra, could be useful in combination with tocilizumab for preventing severe cytokine release and neurotoxicity. In addition, the new specific inhibitors could be effective in mitigating CRS without affecting the cytotoxic efficacy of CAR-T therapy. Conclusion: More lines of research should be opened to elucidate the true implications of these drugs in treating the side effects of CAR-T therapy.

## 1. Introduction

Chimeric antigen receptor T-cell therapy has emerged as an innovative therapeutic strategy for addressing refractory and/or recurrent malignancies, including chemotherapy-resistant acute lymphoblastic leukemia (ALL), chronic lymphocytic leukemia (CLL), and non-Hodgkin lymphoma (NHL) [1,2].

CAR-T therapy is a new form of immunotherapy based on the activation and potentiation of the immune system through the infusion of T cells modified ex vivo to directly attack the tumor. The process consists of extracting the patient’s own lymphocytes by apheresis and genetically modifying them in the laboratory so that they express a specific synthetic receptor, called a “chimeric antigen receptor” (CAR), effective against tumor antigens. These modified lymphocytes subsequently expand and are re-infused into the patient to combine the antigen-binding properties of the antibodies with the effector functions of the lymphocytes [2,3,4].

After leukocytapheresis and before the infusion of the cells, patients who are candidates for therapy should undergo a lymphodeplector chemotherapy regimen based on fludarabine and cyclophosphamide in order to control the disease until infusion and to allow subsequent clonal expansion [5]. The process of CAR-T therapy is represented in Figure 1 [6,7].

Clinical trials showed that CAR-T therapy was able to lead to complete remission or minimal residual disease, as well as prolonging the overall survival and remission time in those hematological diseases for which treatment had been exhausted [8,9,10,11,12,13]. In fact, they showed such good results when targeting the CD19 tumor antigen that several types of drugs have been marketed. One of these, tisagenlecleucel, is primarily intended for older adults with refractory lymphomas, pediatric patients, and young adults with refractory ALL, while axicabtagene ciloleucel (Axicel) is approved for the treatment of diffuse large B-cell lymphoma (DLBCL) and primary mediastinal large B-cell lymphoma. Another drug, brexucabtagene autoleucel (Brexu-cel), has been recently ratified for the treatment of mantle cell lymphoma (MCL) and pediatric acute B-cell lymphatic leukemia (B-ALL). Finally, in February 2021, the use of lisocabtagene maraleucel (Breyanzi) was authorized in the United States for adult patients with relapsed or refractory large B-cell lymphoma after two or more lines of systemic treatment [3,14,15,16].

However, as with any drug, the therapy is not free of side effects. As it is an immunological therapy, when the antigen receptors are introduced into the body, T lymphocytes trigger an immune response that attacks the tumor but can also attack healthy cells. The two most frequent adverse effects of the infusion of CAR-T cells are neurotoxicity syndrome associated with immune effector cells (ICANS) and cytokine release syndrome (CRS) [1,14,17,18,19].

Both syndromes are the consequence of an extensive inflammatory state resulting from T cells releasing inflammatory cytokines and chemokines (such as interleukins (ILs), tumor necrosis factor (TNF), and interferons (IFNs)), which activate dendritic cells and other host T cells. These generate a positive feedback loop that results in the massive destruction of healthy tissue, leading to reversible and widespread organ dysfunction [3,14,16,18,20].

CRS usually appears during the first week after infusion. It is characterized by signs and symptoms such as fever, nausea, headache, myalgia, tachycardia, hypotension, hypoxia, organ dysfunction, cytopenias, coagulopathies, hemophagocytic lymphohistiocytosis, or other renal, gastrointestinal, and hepatic alterations. If any of these symptoms present, early action should be taken to mitigate them. However, in some cases, the presentation is aggravated, leading to life-threatening conditions, such as hypotension at values that require high doses of vasopressors or hypoxia that requires invasive mechanical ventilation [20,21,22].

On the other hand, ICANS can manifest during or after CRS in highly diverse ways, such as encephalopathies, cognitive defects, dysphasias, seizures, and cerebral edema [23,24,25].

In addition, factors such as the tumor burden, the infusion dose, fludarabine lymphodepletion, the existence of concurrent infectious and chronic diseases, and early elevations of cytokines are directly associated with the severity of the syndromes [26,27,28].

Therefore, clinical and analytical monitoring is essential for patients undergoing this therapy. Signs such as hypotension, hyperthermia, tachypnea, tachycardia, and desaturation below basal levels are primary indicators for the detection of CRS [29]. The ICE scale is also a fundamental instrument for the detection of neurotoxicity [30].

In parallel, frequently analyzing the levels of C-reactive protein, ferritin, transaminases, IL-6, IL-1, IL-8, IL-10, IL-15, IFN-γ, and TNF-α can help in the early prediction and diagnosis of CRS and ICANS [31]. Sporadic lumbar punctures for the measurement of GM-CSF, IL-1, IL-8, IL-10, and MCP1 may also be useful in the management of neurotoxicity [32,33].

To address CRS and ICANS, in addition to symptomatic management, different lines of action based on monoclonal and corticoid antibodies have been developed. The first-line and demonstrably most effective treatment is the use of tocilizumab, an IL-6 inhibitor [1,3,13,16,17,32]. However, it has shown little efficacy in preventing and managing ICANS due to its poor ability to cross the BBB [1,17,33].

Recent clinical trials have investigated the role of other interleukin inhibitors, such as anakirna and siltuximab, in preventing and treating both CRS and ICANS [14,17,18,33,34]. The use of interleukin inhibitors is a new and experimental therapy in some medical disciplines. Its mechanism of action, benefits, and disadvantages are not fully understood. Therefore, its use as a treatment in this therapy is an innovation, especially because of the potential benefits it has on the survival of a patient receiving CAR-T therapy.

Thus, the correlation between the elevation of different interleukins and the emergence of CRS and ICANS secondary to CAR-T therapy was the reason behind this study: Analyze the efficacy of the use of interleukin inhibitors to prevent and treat cytokine release syndrome and the neurotoxicity associated with CAR-T therapy. This objective was conceived in the interest of patient safety. Due to the promising character of the therapy in the cure of cancer, research must be expanded to understand and address the side effects of its use. The release of cytokines and neurotoxicity secondary to the use of this therapy can seriously compromise patient health. For this reason, it is essential to adequately understand the mechanism resulting in these side effects, the factors involved, and which methods are safe and effective for addressing them.

## 2. Methods

This article is a literature review that compiles existing information on the efficacy of interleukin inhibitors for combating the toxicities of CAR-T therapy.

Five databases were used for the selection of appropriate articles: PubMed, Scopus, Web of Science, Embase, and ProQuest. All of these databases include articles in which the main topics are health and medical investigations.

First, in all of the databases, a broad search was carried out to estimate the number of articles published on the topic of interest. Next, a more detailed search was carried out, focused on achieving the objective of the study.

The keywords, selected according to the MeSH Terminology, were as follows: “CAR-T cell”, “Cytokine release syndrome”, and “IL Inhibitors” joined with the Boolean AND. English words were used because, during the first search, most of the articles found were in English. For the selection of articles, the titles and summaries of the articles were read, and if this information was insufficient, the articles in question were quickly read.

The inclusion criteria were access to the full-text document, articles published in the last 10 years, experimental studies, and studies carried out on patients with homeopathies and malignant tumors who had undergone CAR-T therapy. All the articles studied that conducted research on humans followed the ethical principles of the Declaration of Helsinki adopted in 1975 and revised in 2013 [35].

The exclusion criteria were studies conducted on animals, studies on patients under 18 years of age, and studies involving patients sick with COVID-19.

Animal studies were excluded from the study because the aim of the review was to study the efficacy of interleukin inhibitors in human patients treated with CAR-T therapy, not in animals. Animal studies would demonstrate results from the primary phases of research on the use of these drugs.

The selection process for each of the databases is represented in a PRISMA flowchart (Figure 2) that was developed with the Cochrane review software Review Manager (RevMan) [36].

## 3. Results

The main results obtained from the analysis of the articles are presented in Table 1.

Multiple interleukin inhibitors that could potentially serve as drugs for the treatment of CRS are described below:

### 3.1. Ertanecept

Zhang et al. discovered through their case study that ertanecept was effective in reducing the levels of TNF-α (*p* = 0.03), IL-6 (*p* = 0.007), and IL-10 (*p* = 0.01) in eight patients undergoing CAR-T therapy and with symptoms associated with CRS [37]. Therefore, ertanecept could be effectively used for the management of CRS associated with CAR-T therapy, especially in patients with high elevations of TNF-α.

### 3.2. Tocilizumab

Regarding tocilizumab, Cheng et al. demonstrated that it was a powerful IL-6 inhibitor effective in reducing the symptoms associated with CRS in four patients undergoing CAR-T therapy. The dose and frequency used depended on the degree of severity and refractoriness of the presentation [38].

Similarly, in a report on 34 cases, Feng et al. observed that the use of tocilizumab did not affect the duration of the effect of CAR-T cells against malignant cells (*p* = 0.040), and therefore, its use did not affect the efficacy of the therapy (*p* = 0.061) [41], being a useful and safe drug.

### 3.3. Tocilizumab in Combination

IL-6 inhibitors are the key drugs for managing moderate or severe CRS. These include tocilizumab and siltuximab, alone or in combination.

The efficacy of these drugs was reaffirmed in the cohort study conducted by Abboud et al. on 75 patients. In this study, it was found that the use of tocilizumab and siltuximab alone or in combination reduced CRS even in its most severe forms [44].

On the other hand, through a comparative study of four patients, Chen et al. also showed that siltuximab could be useful and safe for treating CRS on its own due to its IL-6-inhibitory capacity (*p* = 0.001) [42].

The potential benefits of the combined use of interleukin inhibitors were also demonstrated by the research of Gutierrez et al., which consisted of conducting a questionnaire in 11 centers that performed CAR-T therapy. Their conclusions were as follows: first, that tocilizumab also combats neurotoxicity (82% of centers); second, that the siltuximab–tocilizumab combination could be effective in most refractory cases of CRS (55% of centers); finally, that anakinra could be a satisfactory complement to tocilizumab for the most severe and refractory cases of CRS, as well as for the prevention of ICANS (55% of centers), due to its ability to cross the blood–brain barrier [40].

This last aspect of the usefulness of the combination of tocilizumab and anakinra is supported by the results obtained in the case study by Jatiani et al. showing that tocilizumab combined with anakinra helped to prevent CRS in its most severe stages (in both of the two patients submitted) [39].

Furthermore, Abrams on et al. reported in a multicenter cohort study conducted with 269 patients that tocilizumab alone was useful for treating CRS secondary to CAR-T therapy (50% of patients). However, they propose that the administration of anakinra and siltuximab could be more useful for the most severe or refractory cases of CRS [46].

### 3.4. TO-207

In addition to the well-known tocilizumab, anakinra, and siltuximab, there are other inhibitors, such as TO-207 and ibrutinib. TO-207 is a newly discovered inhibitor of multiple cytokines confirmed in a clinical trial conducted by Futami et al., who discovered its potential role in mitigating ICANS and CRS secondary to CAR-T therapy due to its ability to inhibit the secretion of inflammatory cytokines, such as IL-6, IL-8, IL-18, MCP-1, and TNF-8. It achieves this without affecting the antitumor capacity of CAR-T cells (*p* < 0.001) [45].

### 3.5. Ibrutinib

In contrast to TO-207, ibrutinib did not demonstrate effectiveness against CRS or ICANS. According to Geyer et al., ibrutinib could instead potentiate CRS and ICANS due to its ability to expand CAR-T cells (*p* = 0.04) [43].

## 4. Discussion

CAR-T therapy is an innovative and effective therapeutic option for the treatment of hematological malignancies refractory to previous treatment lines or multiple relapses [1,2,3,4,5,6,7,8,9,10,12,16,17,18,19,20,21,22,23,24,25,26,27,28,29,30,32,34,37,38,39,40,41,42,43,44,45,46].

It has been approved and recognized in many countries for its ability to prolong survival and disease-free periods with an appropriate level of safety. However, some of the adverse effects manifesting after the expansion of CAR-T cells have been a challenge for scientists and health professionals dedicated to the management of patients undergoing therapy [1,3,4,5,6,7,8,9,10,11,12,17,18,23,26,27,29,30,31,39].

The most worrisome adverse reactions to CAR-T therapy are cytokine release syndrome and neurotoxicity. Both derive from an extensive release of cytokines and are characterized by increases in the serum levels of interleukins and other cytokines (such as IL-6, IL-2, IL-8, IL-10, and TNF-α) [1,2,16,20,37,38,39,40,41,42,43,44,45,46].

In all the articles studied, it was observed that the efficacy and usefulness of the different drugs in combating CRS were based on their ability to reduce the levels of the pro-inflammatory cytokines IL-1, IL-2R, IL-6, IL-8, IL-10, IL-18, TNF-α, and MCP-1 [35,36,37,38,39,40,41,42,43,44].

Depending on the level at which the cells are altered, their modification can generate CRS or ICANS of lower (grade 1) or greater severity (grade 4), as shown in Appendix A and Appendix B [47].

The results of this review serve to confirm that tocilizumab is the drug with the greatest efficacy in mitigating the symptoms of CRS grade 1 and 2 due to its ability to reduce the cytokines IL-2R, IL-6, IL-8, IL-10, and TNF-α without decreasing the cytotoxic capacity of CAR-T cells [37,38,39,40,41,44,46]. For this reason, tocilizumab is the only interleukin inhibitor approved for use against CRS caused by first-line CAR-T therapy [48,49].

However, these syndromes are sometimes refractory to the use of tocilizumab, and other lines of support are required [37,38,39,40,41,42,43,44,45,46]. The study by Nagle et al. proposes that corticosteroids are a good treatment with which to complement tocilizumab and, in addition, have significant efficacy against ICANS [48]. However, Brentjens et al. revealed that the use of corticosteroids decreased the efficacy of CAR-T therapy [50].

In addition, Neelapu et al. showed that tocilizumab was only useful for treating CRS and neurotoxicity if the latter was secondary to CRS [51]. Furthermore, Nishimoto et al. demonstrated that its use for the treatment of isolated ICANS was ineffective because its molecular size prevents it from crossing the blood–brain barrier, and it can only inhibit serum IL-6, consequently increasing the concentration of IL-6 in the cerebrospinal fluid and aggravating the situation [52].

To solve this problem, some scientists have investigated other interleukin inhibitors that may be useful in mitigating CRS and ICANS without affecting the efficacy of the therapy [44,53,54,55].

In our research, some drugs, such as siltuximab or anakinra, showed statistically significant results in the mitigation of CRS grade 3 and 4 induced by CAR-T therapy when used after treatment with tocilizumab, decreasing the need for subsequent doses of this drug or the need to use corticosteroids [37,40,42,44,46].

The use of siltuximab after a patient shows resistance to tocilizumab is considered a good therapeutic alternative to steroid due to siltuximab’s strong ability to reduce the levels of IL-6 and sIL-6R [40,42,44,46].

Parallel to this research, other studies, such as those carried out by Rivera et al. and Acharya et al., also show that the combined use of siltuximab and tocilizumab is very effective in improving the symptoms of CRS and preventing and treating neurotoxicity [54,55]. However, the work of Neelapu et al. considers that the use of both drugs is only useful for the initial management of CRS in adults and that anti-IL-6 therapy is ineffective in treating neurotoxicity [56].

Therefore, interest has arisen in anakinra, an IL-1 inhibitor used especially in arthritic diseases. The selected studies demonstrate the ability of anakinra to act not only on CRS but also on ICANS induced by CAR-T therapy. Its effects are due to its ability to reduce IL-1 levels and penetrate the blood–brain barrier to act on neurotoxicity [40]. In addition, the results of the review show that its use after tocilizumab decreases the need for extra doses of tocilizumab and steroid use [39,40,46].

Articles such as that published by Riegler et al. also affirm that anakinra is a good option not only for the treatment of ICANS but also for preventing and reducing the mortality associated with CRS [57,58].

This review also shows that TNF-α inhibitors, such as ertanecept, have greater efficacy than tocilizumab in lowering the levels of TNF-α and, therefore, in improving some of the symptoms, such as fever, hypotension, and arthralgia, without producing side effects [37,38]. Therefore, ertanecept may be a good substitute for tocilizumab for mitigating CRS [38].

Moreover, the studies by Lee et al. and Grupp et al. propose the combination of tocilizumab with ertanecept and/or corticosteroids in short courses as promising treatment options for mitigating severe CRS [59,60,61].

To expand the therapeutic options, TO-207, a multi-cytokine inhibitor, has been developed and is able to prevent and mitigate CRS. This new inhibitor could be more effective than tocilizumab in decreasing the secretion of pro-inflammatory cytokines due to its specific action on monocytes, its lack of effect on CAR-T cells, and, thus, its prevention of the development of CRS without altering the efficacy of therapy [45].

There have not been many studies in which TO-207 has been used in CAR-T therapy. Studies such as those conducted by Uesato et al. and Kakegawa et al. demonstrate that TO-207 improves the symptoms associated with CRS and ICANS and acts with CAR-T cells to increase antitumor efficacy and increase the survival times of patients who receive it [45,62,63].

However, the use of anticytokines is not always safe. Ibrutinib is a protein kinase inhibitor that some patients may take prior to CAR-T therapy. From our research, it can be concluded that this drug increases the in vivo expansion of CAR-T cells and improves the tumor response to treatment to a statistically significant degree. It also, however, predisposes the patient to the development of CRS greater than grade 2, which must be treated with tocilizumab [43].

The study by Fan et al. states that ibrutinib does not increase pro-inflammatory interleukins and that it is a good IL-1-receptor inhibitor [64]. In fact, Fraietta et al. and Turtle et al. support the proposal that the use of ibrutinib supplemented with the CAR-T cell production process could achieve significantly greater expansion of T cells and could also achieve significantly higher CAR-T cell yields without any CRS [65,66,67].

It should be noted that, although there were few articles found in relation to CAR-T therapy and the use of interleukin inhibitors in hematological neoplasms, articles focusing on this treatment in solid tumors were even scarcer. This is because CAR-T therapy in cancer neoplasms is still under study, and few trials have been conducted in humans. The reason for this is that, in the tumor microenvironment, the lack of specific target antigens and inactivation by control inhibitors hamper the durability of CAR-T cells, impair their ability to penetrate the tumor, and increase the severity of side effects, such as CRS and neurotoxicity. Therefore, the research focuses on studying targets or therapeutic combinations of CAR-T cells (such as NK cells) that would decrease toxicity, increase the antitumor capacity of the cells, and prolong the disease-free time of patients who undergo the treatment [68,69,70].

As can be seen, there is a wide range of articles and studies dedicated to investigating new therapies to address the neurotoxicity and CRS of CAR-T therapy. However, they are clinical trials or in vitro studies in which the drugs are not used in large populations. This is related to the fact that, although interleukin inhibitors are accepted and approved for inflammatory diseases, their anti-inflammatory activity in chimeric antigen receptor therapy is not yet known. However, they are considered potential lines of treatment to address the adverse effects of CAR-T therapy.

This new approach has especially enhanced the treatment of CRS in patients infected with SARS-CoV-2. Many of the articles found in the initial search during this study addressed the use of interleukin inhibitors to effectively treat the severe cytosine cascade derived from the inflammatory period of infection [71,72,73,74].

Due to the lack of consensus and clarifying results in the studies found, more research should be undertaken with regard to other interleukin inhibitors that may be effective in reducing or replacing doses of corticosteroids or tocilizumab. That is, it is important to focus research on clarifying the mechanisms of action and activity of siltuximab, anakinra, ertanecept, and TO-207 to address the CRS and ICANS derived from CAR-T therapy in order to clarify a therapeutic and administration plan and to achieve their approval for use in CAR-T therapy.

This review has had some limitations. CAR-T therapy is a relatively new line of anticancer treatment that causes many aspects of the pathophysiology associated with CRS and ICANS, for which the best options for mitigation are not clearly understood. This aspect significantly restricted the scope of this review. Firstly, because the number of articles investigating the use of interleukin inhibitors was very small, we found some experimental articles about the use of corticoids or Tocilizumab, but few of them were on other inhibitors. Secondly, because many of the drugs of interest have only been used experimentally in animals or have been used for the paediatric population also eligible for this treatment. However, this review expands and contrasts the information of the selected articles.

## 5. Conclusions

Based on the research reviewed in this paper, it can be concluded that the most widely used interleukin inhibitor is tocilizumab. This is because tocilizumab is effective in mitigating the symptoms of CRS.

For addressing tocilizumab-refractory CRS and associated neurotoxicity, siltuximab may be an effective complementary therapy.

The IL-1 inhibitor anakinra may be used to complement tocilizumab in preventing and treating ICANS, as well as in decreasing the severity of CRS.

Other drugs, such as TNF-α inhibitors, may also be useful alone or in combination with tocilizumab to address the side effects of CAR-T therapy without affecting the antitumor efficacy of cells with the chimeric antigen receptor.

It is unclear whether the use of ibrutinib has an adequate mechanism of action for treating CRS, although it does increase the efficacy of CAR-T therapy.

Further lines of research should focus on the use of known interleukin inhibitors to address the adverse effects derived from the expansion of CAR-T cells because they have been shown to be safe and useful immunomodulators and do not affect antitumor efficacy.

## Figures and Tables

**Figure 1 diseases-10-00041-f001:**
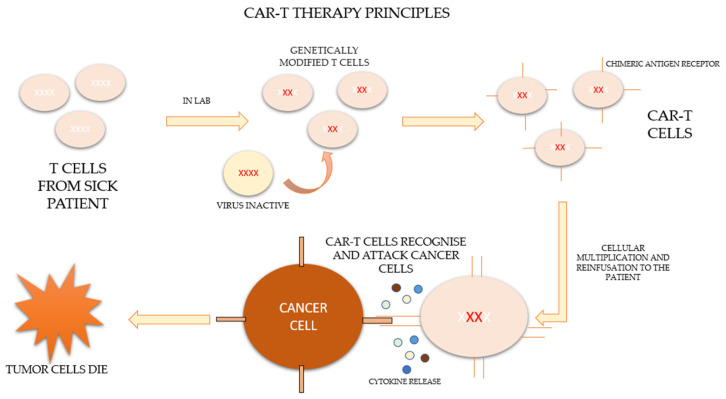
Principles of CAR-T therapy.

**Figure 2 diseases-10-00041-f002:**
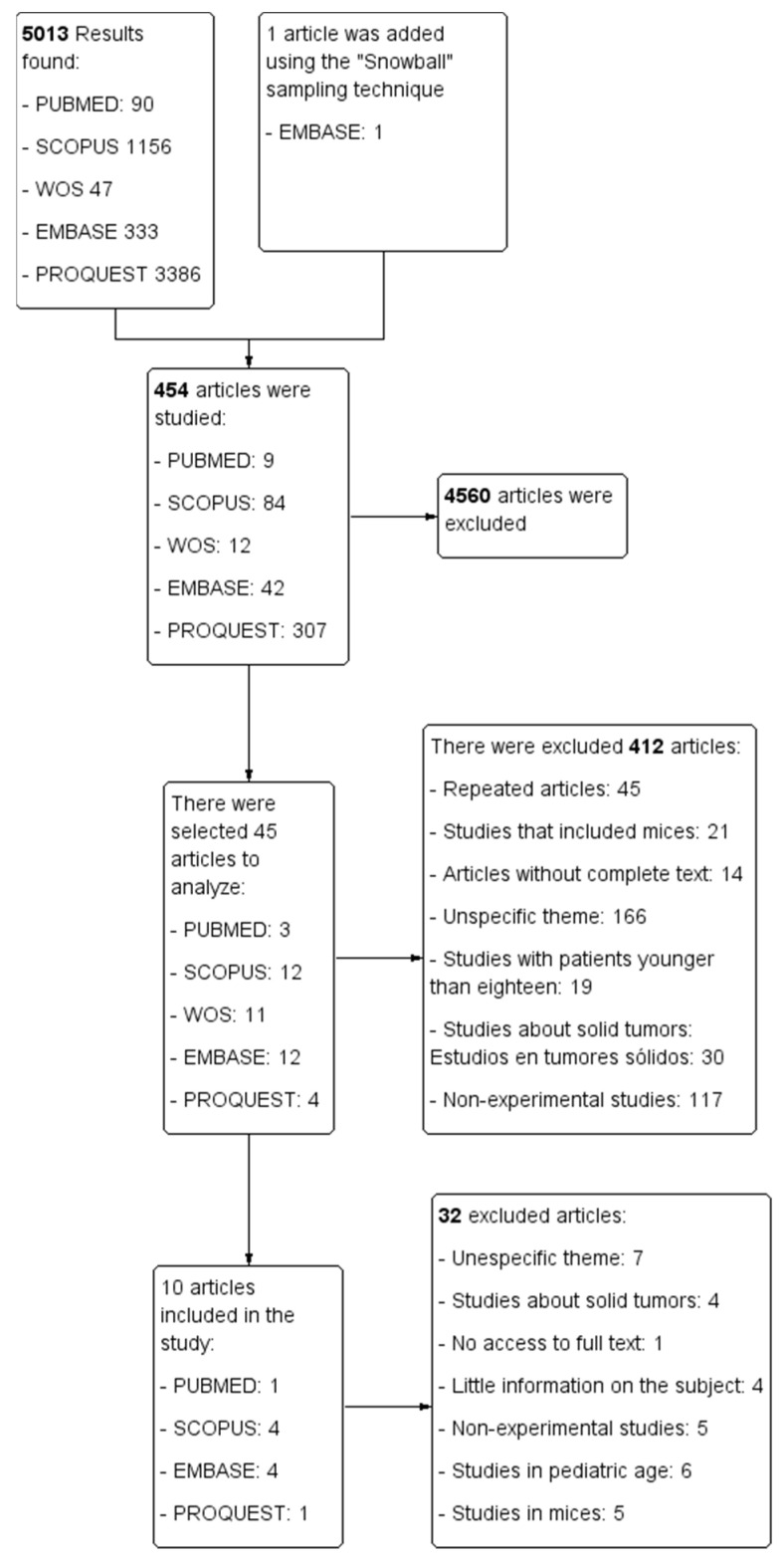
PRISMA flowchart.

**Table 1 diseases-10-00041-t001:** Summary of the results obtained after the study of the articles.

First Author	Studio Design	Sample Size (n)	Drug	Main Results
Zhang et al. [37]	Case report	8 patients (4 men and 4 women)	Ertanecept	Patients who presented CRS after CAR-T infusion therapy were treated with ertanecept, which mitigated the fever, hypotension, and arthralgia due to decreases in TNF-α. (* *p* = 0.03), IL-6 (* *p* = 0.007), and IL-10 (* *p* = 0.01) Ertanecept, a TNF-α inhibitor, is able to resolve CRS without causing side effects or decreasing the effectiveness of CAR-T therapy (*p* > 0.05).
Chen et al. [38]	Case report	4 patients (3 men and 1 woman)	Tocilizumab	Tocilizumab was able to reduce the altered levels of IL-2R, IL-6, IL-8, IL-10, and TNF-α in 2 of the 4 patients. The use of tocilizumab at low doses (4 mg/kg) may be useful for patients with CRS of mild or moderate severity. The use of tocilizumab at high doses (8 mg/kg) would be indicated in patients suffering from persistent fever, hypotensive shock, acute respiratory failure, and rapid progression of LCHS. Even if symptoms do not improve within 8–24 h, another dose of tocilizumab may be given (2 of 4 patients).
Jatiani et al. [39]	Case report	2 patients (2 men)	Anakinra and tocilizumab	The administration of anakinra as anti-IL-1R therapy limits the development and duration of CRS when administered as an adjunct to tocilizumab and decreases the need for additional doses of tocilizumab or steroids (immunosuppressant) (1 of 2 patients).
Gutierrez et al. [40]	Cutting studio	11 centers Unknown patients	Anakinra, siltuximab, and tocilizumab	Among the 11 hospitals that made up the study, the specific practices performed for the treatment of toxicities included the use of tocilizumab to treat CRS and neurotoxicity with symptoms of CRS (82% of centers; n = 9). The use of siltuximab for grade 3 or 4 CRS refractory to tocilizumab (55% of centers; n = 6) The use of anakinra in CRS grade 3 or 4, refractory to tocilizumab, and in neurotoxicity (55% of centers; n = 6).
Feng et al. [41]	Cohort study	89 patients (48 men and 41 women)	Tocilizumab	Treatment with tocilizumab allows CAR-T cells to be safely administered in all and without compromising the efficacy of therapy (*p* = 0.061). Treatment with tocilizumab (0.377; 95% CI 0.001–0.033; * *p* = 0.040) is considered an independent risk factor associated with total cost during CAR-T therapy due to its use with CRS.
Chen et al. [42]	Comparative study	4 patients (1 male and 3 female)	Siltuximab and tocilizumab	The addition of siltuximab reduces IL-6 concentrations by 56% to 74% in patients undergoing CAR-T therapy (* *p* = 0.001). Tocilizumab caused a small but statistically significant reduction in sIL-6R concentrations to between 51% and 70%.
Geyer et al. [43]	Clinical trial	20 patients (14 men and 6 women)	Ibrutinib and tocilizumab	Ibrutinib causes an increase in the levels of IL-6 (* *p* = 0.01) and IL-10 (* *p* = 0.02) that predisposes the patient to the appearance of CRS of grade 2 or higher. Take ibrutinib prior to CAR-T therapy predisposes the patient to further cell expansion. (*p* = 0.040) 80% of patients with CLL who took ibrutinib had objective responses after CAR-T therapy. Of the 5 patients who received ibrutinib prior to the infusion of CAR-T therapy, 3 required tocilizumab to reverse grade 2 or 3 CRS.
Abboud et al. [44]	Cohort study	75 patients (40 men and 35 women)	Siltuximab and tocilizumab	7 patients with CRS were treated with tocilizumab. Of these, 100% resolved signs and symptoms within 48 h, CRP levels fell below 50% of the peak values, and 86% (6 of 7) survived for 100+ days, including those with severe CRS. IL-6 is a key mediator for CRS, so tocilizumab and siltuximab are possible effective therapeutic approaches.
Futami et al. [45]	Clinical trial	In vitro study. Unknown volunteers	TO-207	New multi-cytosine inhibitor TO-207 has modest effects on cytokine secretion in CAR-T cells (* *p* < 0.001) TO-207 specifically inhibits pro-inflammatory cytokines of monocytes (IL-6, IL-8, IL-18, MCP-1, and TNF-α) without affecting cytokine production, or CAR-T cell efficacy
Abramson et al. [46]	Multicenter multi-cohort study	269 patients (174 men and 95 women)	Anakinra, siltuximab, and tocilizumab	Tocilizumab was useful in treating LSS in 53 patients (50% of those with CRS). Siltuximab and anakinra may be effective as a treatment for grade 4 CRS (used in the only patient who had grade 4 CRS).

* Results are considered to be statistically significant when the *p*-value is less than 0.05.

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
