# Peer review of "Interleukin Inhibitors in Cytokine Release Syndrome and Neurotoxicity Secondary to CAR-T Therapy"

_diseases, 2022, doi:10.3390/diseases10030041_

Round 1

Reviewer 1 Report

The revised version of the review manuscript by Ferreros and Trapero entitled “Interleukin Inhibitors in Cytokine Release Syndrome and Neurotoxicity Secondary to CAR-T Therapy” has addressed the points of criticism raised in a satisfactory way.

Author Response

We again apreciate the editor’s comments and now hope that it can be considered for publication

Reviewer 2 Report

The authors investigated about the efficacy of interleukin inhibitors in cytokine release syndrome (CRS) and neurotoxicity secondary to CAR-T therapy. The review is clearly written, its original and of interest in its field. 

I recommend that the review be accepted with minor revision:

a)     The authors should better expleined the rational behind this paper

b)    The authors should provide the limitation of this review.

c)     The literature is poor exhaustive. The authors should add more references. 

Author Response

Point 1 “The authors should better expleined the rational behind this paper”

We have added: “The use of interleukin inhibitor is already a new and experimental therapy in some medicine discipline. Its mechanism of actuation, benefits and disadvantages aren’t totally known. For this reason, the use of them as a treatment in this therapy suppose an innovation, especially because the potentially benefits that it has in the survival of a patient who receive the CAR-T teraphy.”

Point 2. The authors should provide the limitation of this review.

 We have modified the conclusion, we have written some lines which talk about the limitation of this review 

Point 3. The literature is poor exhaustive. The authors should add more references. 

 We have added some bibliography such as the references: 4,6,19,25,60,66,68,73

This manuscript is a resubmission of an earlier submission. The following is a list of the peer review reports and author responses from that submission.

Round 1

Reviewer 1 Report

Puri Ferreros Gómez's review is very interesting and could have a good impact on the scientific community.
Only a few changes are needed. 

Authors should better highlight the purpose of this review and should correct typos and grammatical errors

Why were animal studies excluded? Are there no good animal models of CAR-T therapy of cancer? Please provide a rationale for this.

Author Response

Point 1. Authors should better highlight the purpose of this review and should correct typos and grammatical errors.

Response 1.We thank the reviewers' comments. We have written a little text in which is explained the reason and the importance of this study after the objective. We also have corrected the grammatical errors.  

Reviewer 2 Report

The review by Ferreros and Trapero entitled “Interleukin Inhibitors in Cytokine Release Syndrome and Neurotoxicity Secondary to CAR-T Therapy” proposes to study the efficacy of interleukin inhibitors to address cytokine release syndrome and neurotoxicity secondary to CAR-T therapy. Although the manuscript contains important insights, some points regarding abbreviations, materials and methods, and the discussion, all detailed below need to be improved.

  1. Abstract: “Introduction: CAR-T therapy appears as an innovative therapeutic option to address some recurrent or refractory hematological malignancies.” Please do not use any abbreviations (CAR-T) that were not introduced in the abstract.
  2. Abstract: “Objective: To study the efficacy of interleukin inhibitors to address SLC and neurotoxicity secondary to CAR-T therapy.” Please do not use any abbreviations (SLC) that were not introduced in the abstract. Also, would it not make more sense to use CRS as an abbreviation for cytokine release syndrome? Would SLC be the Spanish abbreviation?
  3. Abstract: “Methodology: Bibliographic review in which 10 quantitative and experimental articles were analyzed, which included cut-off studies, case reports and clinical trials with 11 cancer centers and up to 475 patients over 18 years of age.” It is not quite clear to me what the authors meant by quantitative and experimental. Please explain in the Materials and Methods section.
  4. Introduction: It would be helpful to have a figure explaining the principles of CAR-T therapy.
  5. Introduction, lines 52-55: “And finally, last February 2021, the use of Lisocabtagene maraleucel (Breyanzi) was authorized for adult patients with relapsed or refractory large B-cell lymphoma, after two or more lines of systemic treatment. [3,11,12]” Please, specify in which countries this is authorized.
  6. Introduction, lines 65: “…cytokines and chemokines (IL, TFN, IFN...) …” Please, spell out abbreviation the first time they are used.
  7. Introduction, line 88: “Signs such as blood pressure, body temperature, respiratory rate, sat02 …” Blood pressure, body temperature, respiratory rate, sat02 are not signs as such, but bodily variables. However, hypotension, hyperthermia, etc., are signs. Please, correct.
  8. Introduction, line 92: Please, write out PCR.
  9. Materials and methods: lines 115-117: “To search for this information, 5 databases chosen for their fundamental scientific-health typology were used for the selection of appropriate articles for the development of research. These include Pubmed, Scopus, Web of Science, Embase and Proquest.” I have no idea what “fundamental scientific-health typology” means, please use a different phrasing. Please, also indicate in what these different databases were complementary.
  10. Materials and methods: lines 130-131: “From this information, it was established whether the articles had the appropriate characteristics to be included in the study, or, on the contrary, to be excluded.” This is unclear and can seem subjective. Please, be more specific about the characteristics mentioned.
  11. Material and methods: lines 138-140: “In parallel, among the criteria selected to exclude the articles from the study were: Studies conducted on animals, studies on patients under 18 years of age and with patients sick with COVID.” Why were animal studies excluded? Are there no good animal models of CAR-T therapy of cancer? Please provide a rationale for this.
  12. Results: Please add some text to the results section summarizing and referring to table 1.
  13. Results: Not all table lines indicate the number of patients (for example Guitierrez et al. [34]). Please provide these numbers.
  14. Discussion; the discussion needs to be much more synthetic.
  15. Discussion, lines 254-273 need to be integrated better with the rest of the discussion.
  16. Discussion, lines 280-283: The work of Norelli discussed concerns a mouse model. But animal studies were excluded from the authors’ analysis. So why was work obtained in mice then discussed. Please be coherent on this point and justify why.
  17. Conclusions: lines 355-356: “From all the research it can be concluded that the only interleukin inhibitor approved for use in CAR-T therapy is Tocilizumab.” We do not need research to know that a drug is approved. This is a regulatory decision. Please, rephrase.

Author Response

Point 1. Abstract: “Introduction: CAR-T therapy appears as an innovative therapeutic option to address some recurrent or refractory hematological malignancies.” Please do not use any abbreviations (CAR-T) that were not introduced in the abstract. 

Point 2. Abstract: “Objective: To study the efficacy of interleukin inhibitors to address SLC and neurotoxicity secondary to CAR-T therapy.” Please do not use any abbreviations (SLC) that were not introduced in the abstract. Also, would it not make more sense to use CRS as an abbreviation for cytokine release syndrome? Would SLC be the Spanish abbreviation? 

Response 1 and 2. We thank the reviewer’s suggestion to improve this section, we have added the word before the abbreviation. We also have changed “SLC” by “CRS” in the document. 

Point 3. Abstract: “Methodology: Bibliographic review in which 10 quantitative and experimental articles were analyzed, which included cut-off studies, case reports and clinical trials with 11 cancer centers and up to 475 patients over 18 years of age.” It is not quite clear to me what the authors meant by quantitative and experimental. Please explain in the Materials and Methods section. 

Response 3. We thank the reviewer’s suggestion; we have eliminated this phrase and the papers are defined in the Materials and Methods section.  

Point 4. Introduction: It would be helpful to have a figure explaining the principles of CAR-T therapy. 

Response 4. We thank the reviewer’s suggestion. We have added the image 1 in the document.

Point 5. Introduction, lines 52-55: “And finally, last February 2021, the use of Lisocabtagene maraleucel (Breyanzi) was authorized for adult patients with relapsed or refractory large B-cell lymphoma, after two or more lines of systemic treatment. [3,11,12]” Please, specify in which countries this is authorized.

Response 5. We thank the reviewer’s suggestion. We have added the continent where its use is approved (All UUEE).

Point 6. Introduction, lines 65: “…cytokines and chemokines (IL, TFN, IFN...) …” Please, spell out abbreviation the first time they are used.

Response 6. We thank the reviewer’s suggestion to improve this section, we have added the word before the abbreviation. 

Point 7. Introduction, line 88: “Signs such as blood pressure, body temperature, respiratory rate, sat02 …” Blood pressure, body temperature, respiratory rate, sat02 are not signs as such, but bodily variables. However, hypotension, hyperthermia, etc., are signs. Please, correct.

Response 7. We thank the reviewer’s suggestion. We have written the specific signs that a patient can manifest. 

Point 8. Introduction, line 92: Please, write out PCR.

Response 8. We thank the reviewer’s suggestion we have change it by C-reactive protein

Point 9. Materials and methods: lines 115-117: “To search for this information, 5 databases chosen for their fundamental scientific-health typology were used for the selection of appropriate articles for the development of research. These include Pubmed, Scopus, Web of Science, Embase and Proquest.” I have no idea what “fundamental scientific-health typology” means, please use a different phrasing. Please, also indicate in what these different databases were complementary

Response 9. We thank the reviewer’s suggestion. We have rewritten the phrase to be more coherent and more specific about what databases we have used and the reason why. We did not use any complementary databases.

Point 10. Materials and methods: lines 130-131: “From this information, it was established whether the articles had the appropriate characteristics to be included in the study, or, on the contrary, to be excluded.” This is unclear and can seem subjective. Please, be more specific about the characteristics mentioned.

Response 10. We thank the reviewer’s suggestion. To improve in this section, we have considered that this phrase wasn’t coherent, so we have decided to delete it. 

Point 11. Material and methods: lines 138-140: “In parallel, among the criteria selected to exclude the articles from the study were: Studies conducted on animals, studies on patients under 18 years of age and with patients sick with COVID.” Why were animal studies excluded? Are there no good animal models of CAR-T therapy of cancer? Please provide a rationale for this.

Response 11. We thank the reviewer’s suggestion. In order to be more rationale, we have explained in the text, the reason why we haven’t included the studies which included animals. 

Point 12. Results: Please add some text to the results section summarizing and referring to table 1.

Response 12. We thank the reviewer’s suggestion. To improve this section, we have added a text in which are
synthesized and explained the results of the table 1

Point 13. Results: Not all table lines indicate the number of patients (for example Guitierrez et al. [34]). Please provide these numbers

Response 13. We thank the reviewer’s suggestion. Unfortunately, the number of patients who participate in the study is not specificized in that paper, so “Unknown patients” have been added in table 1. 

Point 14. Discussion; the discussion needs to be much more synthetic.

Response 14. We thank the reviewer’s suggestion. We have tried to remove some phrases and modify and collect some paragraphs to be more synthetics.

Point 15. Discussion, lines 254-273 need to be integrated better with the rest of the discussion.

Response 15. We thank the reviewer’s suggestion. We have changed the way is written this part of text to be more integrated. 

Point 16. Discussion, lines 280-283: The work of Norelli discussed concerns a mouse model. But animal studies were excluded from the authors’ analysis. So why was work obtained in mice then discussed. Please be coherent on this point and justify why.

Response 16. We thank the reviewer’s suggestion. In this section, we have removed this part of the text to avoid confusion.

Point 17. Conclusions: lines 355-356: “From all the research it can be concluded that the only interleukin inhibitor approved for use in CAR-T therapy is Tocilizumab.” We do not need research to know that a drug is approved. This is a regulatory decision. Please, rephrase.

Response 17. We thank the reviewer’s suggestion. To improve this section; We have rephase this part of the text to be more specific.  
